# Reducing mosquito-borne disease transmission to humans: A systematic review of cluster randomised controlled studies that assess interventions other than non-targeted insecticide

**Jane Oliver**[1]*, **Stuart Larsen**[1], **Tim P. Stinear**[1], **Ary Hoffmann**[2], **Simon Crouch**[3], **Katherine B. Gibney**[1,4,5]

**1** The Peter Doherty Institute for Infection and Immunity, University of Melbourne, Melbourne, Australia, **2** Pest and Environmental Adaptation Research Group, Bio21 Institute and the School of BioSciences, University of Melbourne, Melbourne, Australia, **3** Department of Health, Melbourne, Melbourne, Australia, **4** The Royal Melbourne Hospital, Department of Infectious Diseases, Melbourne, Australia, **5** Melbourne Health,Victorian Infectious Diseases Service, Melbourne, Australia

* jane.oliver@unimelb.edu.au

## Abstract

### Background

Mosquito control interventions are widely used to reduce mosquito-borne diseases. It is unclear what combination of interventions are most effective in reducing human disease. A novel intervention study for Buruli ulcer targeting mosquito vectors was proposed for a Buruli ulcer-endemic area of Victoria, Australia. The local community expressed a preference for avoiding widespread residual spraying of pyrethroids. To inform the design of a future cluster randomised control study (cRCT) for Buruli ulcer prevention in Victoria, we conducted a systematic literature review.

### Aims

The aim was to describe cRCT designs which investigated interventions other than non-targeted insecticide for reducing mosquito-borne disease transmission, and comment on the strengths and weaknesses of these study designs.

### Methods

Five medical research databases were searched for eligible literature from the earliest available sources up to 5 July 2019 (Medline, Embase, Web of Science, EBM Reviews, CAB Direct). Reference lists of identified studies were hand searched. Eligible studies were cRCTs using targeted chemical or biological mosquito control interventions, or mosquito breeding source reduction, with the occurrence of mosquito-borne disease as an outcome.

**Data Availability Statement:** All relevant data are within the manuscript and its Supporting Information files.

**Funding:** The authors received no specific funding for this work.

**Competing interests:** The authors have declared that no competing interests exist.

## Results

Eight eligible cRCTs, conducted between 1994–2013 were identified in a variety of settings in the Americas and Asia. Interventions to reduce dengue transmission were mass adult trapping and source reduction. Interventions to reduce malaria transmission were largescale larvicide administration and (topical and spatial) repellent use. Three studies showed the intervention was associated with statistically significant reductions in the disease of interest and entomological indicators. High community engagement with the intervention were common to all three. In two studies, large buffer zones reduced contamination between study arms. Heterogeneity was reduced through increasing study cluster numbers, cluster matching and randomisation.

## Conclusion

High community engagement is vital for a cRCT reducing mosquito-borne disease with a mosquito control intervention. These findings support a mosquito breeding source reduction intervention for *Aedes* control in a future study of Buruli ulcer prevention if local communities are supportive and very engaged. Regular administration of larvicide to sites unsuited to source reduction may supplement the intervention.

### Author summary

Mosquito control interventions are widely used to reduce mosquito-borne diseases, but it is unclear what combination of interventions are most effective in reducing human disease. Given the wide range of mosquito species and the diseases they transmit, different interventions strategies have been implemented across many regions globally, with varying degrees of success. This literature review identified three intervention studies which did not include non-targeted use of insecticide and were associated with statistically significant reductions in the disease of interest and in entomological indicators following the intervention. High community engagement is vital for the success of a cluster randomised control study aiming to reduce mosquito-borne disease with a mosquito control intervention, such as breeding source reduction for *Aedes* control. In two studies, large buffer zones reduced contamination between study intervention and control arms. Differences between the study arms were reduced through increasing study cluster numbers, cluster matching and randomisation. Regular administration of larvicide to potential breeding sites that are unsuitable for source reduction may supplement this intervention strategy.

## Introduction

Mosquito control interventions are vital for the suppression of mosquito-borne diseases. Given the diversity in mosquito species and in the diseases they transmit, different interventions strategies have been implemented across many regions globally, with varying degrees of success.[1–6] Such interventions may aim to reduce mosquito blood-feeding on humans by repelling mosquitoes, or may physically block mosquitoes from reaching a blood meal (e.g. bed nets). Some interventions seek to reduce the local mosquito abundance (and subsequent blood-feeding) by disrupting the mosquito lifecycle.[7] To illustrate, 'source reduction'

removes suitable sites for mosquito breeding such as stagnant water.[8] Larvicides are juvenile insect hormone analogues which inhibit mosquito larvae development in water sources. Adulticides reduce adult mosquito numbers. Mosquito sterilisation can reduce the population abundance, as may trapping of adults and/or larvae, and introducing and/or promoting predator populations.[7] Reports of mosquito control intervention studies in areas with endemic mosquito-borne disease are widely available in the scientific literature, yet how such interventions affect disease incidence in local human populations is often not reported. A review investigating the effectiveness of different control measures for reducing *Aedes aegypti* proliferation concluded that governments relying on chemical controls should consider adding community mobilisation to their prevention efforts; however, clinical endpoints were not considered.[9]

Buruli ulcer is an infection of subcutaneous tissue caused by *Mycobacterium ulcerans*.[10] An epidemic is currently occurring in Victoria, Australia.[11,12,13] Most recent Victorian cases are related to an endemic focus on the Mornington Peninsula—a popular tourist destination.[14–16] A growing body of evidence implicates biting insects as having a key role in the transmission of *M. ulcerans* in Australia.[17,18]. The Mornington Peninsula epidemic may be mediated by environmentally contaminated mosquito vectors (in particular *Aedes notoscriptus*; a species of freshwater container breeder mosquitoes common in suburban areas[19] on which *M. ulcerans* has been detected in a Buruli ulcer endemic area in Victoria[17]).[18]

An intervention study for Buruli ulcer targeting arthropod vectors has, to our knowledge, never been undertaken. Any mosquito control intervention study that aims to reduce mosquito-borne disease rates in humans needs to consider clustering of study groups to account for mosquito ecology.[20] Following community engagement in the Mornington Peninsula regarding a proposed intervention study targeting mosquitoes to reduce Buruli ulcer incidence, local government and some residents expressed a preference for interventions that did not involve widespread residual spraying using pyrethroids due to concerns around perceived health effects and possible collateral damage to other insect populations. Consequently, this literature review was developed to inform the design of a future cluster randomised control study (cRCT) aiming to reduce Buruli ulcer transmission using mosquito control intervention(s) other than non-targeted insecticide spraying. The aim of this study was to review cRCT designs used to investigate interventions other than non-targeted insecticide for reducing mosquito-borne disease transmission to humans, and comment on the strengths and weaknesses of these study designs.

## Methods

### Information sources

Our study conforms to the Preferred Reporting Items for Systematic Reviews and Meta-analysis (PRISMA) guidelines.[21] We retrieved literature from Ovid Medline; Embase Classic+-Embase; Web of Science; EBM Reviews—Cochrane Central Register of Controlled Trials; and CAB Direct in July 2019.

### Search strategy

Search strategies for each database were developed with assistance from Medical Reference Librarians at the University of Melbourne. The search terms included: "'control*' or 'prevent*', 'cluster' and 'random*', 'rct*', 'mosquito*' or 'culicidae' or 'malaria*' or 'dengue' or 'arbovir*' or 'Buruli' or 'aedes' or 'malaria' or 'West Nile' or 'chikungunya' OR 'yellow fever' or 'filariasis' or 'tularemia' or 'dirofilariasis' or 'Japanese encephalitis' or 'Saint Louis encephalitis' or 'Ross River' or 'Barmah Forest' or 'la crosse' or 'zika' or 'keystone', 'bed net*'or 'pyrethrum*' or 'pyrethroid' or 'pesticide*' or 'repellent*' or 'spray*' or 'chemical*' or 'retardant*' or 'coil*' or

'community mobili*' or 'water' or 'source reduction' or 'trap*' or 'biocontrol' or 'larvicid*' or 'adulticid*' or 'steril*' or 'tetracycline' or 'oil drip*' or 'DDT' or 'lindane' or 'malathion' or 'propoxur'. Detailed search strategies with key terms used for each database are presented in S1 Appendix.

To identify additional studies, we reviewed and hand searched reference lists of identified literature reviews and meta-analyses. Citations of studies thought potentially eligible for inclusion were extracted for screening, with duplicates removed and the remaining articles screened.

## Eligibility criteria

Articles were eligible for inclusion in this review if the following criteria were met:

- Online abstract available.

- Written in English.

- Published in a peer reviewed journal at any time prior to July 2019 (not including conference proceedings).

- Described completed research (research protocols were excluded).

- cRCT design.

- The intervention activity, which was applied only to the intervention area, targeted mosquitoes without non-targeted use of insecticide.

- Quantified the occurrence of mosquito-borne disease in the study population according to control/intervention status following the intervention.

When non-targeted insecticide/s was applied in both the control and intervention arms using the same (or a very similar) approach, and a separate intervention was applied, the study was eligible for inclusion. This assumes the effects of the non-targeted insecticide were balanced across both study arms, and that any difference in outcomes between study arms is attributable to the intervention activity which was applied only in the intervention area. Where non-targeted insecticide/s was applied outside of the study protocol (such as through a government-initiated mosquito control programme), the study was eligible for inclusion and the effects of such activities were considered as possible sources of contamination bias.

## Study selection

Two authors (JO and SL) screened the titles and abstracts. Articles were evaluated for inclusion according to the eligibility criteria. Where it was unclear if an article met the inclusion criteria based on the title and abstract, or where the two reviewers disagreed, the full text of the manuscript was reviewed and a decision regarding article eligibility was made by the first author (this process occurred for approximately one-quarter of screened articles).

## Data abstraction

The full text of included articles was reviewed, with data abstracted independently by the two reviewers using a template specifying relevant data fields. Inconsistencies in abstracted data were resolved by re-checking the article, with discussion and consensus within the study team. Data were abstracted to a data collection template (S2 Appendix), including the study citation; disease and vector; study setting and date; inclusion/exclusion criteria; intervention(s); randomisation method, data collection and analysis methods; number, level (household, village,

region etc.) and distribution of clusters; number and age distribution of participants; outcomes (entomological indicators of vector breeding and occurrence of mosquito-borne disease cases in the control and intervention arms); loss to follow-up, assessment of bias and study limitations.

### Outcomes and prioritisation

Data around the study setting, design, analysis and effectiveness were used to address the first objective: to describe cRCT designs used to investigate interventions other than non-targeted insecticide aiming to reduce transmission of mosquito-borne disease in humans. Data on the effectiveness of the outcome and assessments of the studies' strengths, limitations and bias were used to address the second objective: to comment on the strengths and weaknesses of included study designs.

### Bias assessment

Both reviewers independently assessed, and commented on, the risk of bias, strengths and limitations of included studies. When assessing the risk of bias in individual studies, the Joanna Briggs Institute Critical Appraisal Checklist for Randomized Controlled Trials tool was used (http://joannabriggs.org/research/critical-appraisal-tools.html). Inconsistencies between reviewers were addressed as previously described.

### Synthesis of results

Overall results are described qualitatively, with emphasis on the types of limitations associated with different study designs. Simple counts and short descriptions are presented, in particular when describing intervention types, study designs, study settings, analysis methods (Table 1), strengths and limitations (Table 2).

## Results

### Article searches and screening

Following searches of the five medical research databases, 1,471 article citations were identified of which 1080 were duplicates and 391 articles underwent title and abstract screening (Fig 1). Twenty-one literature reviews or meta-analyses were identified; searching their reference lists identified an additional 67 articles for title and abstract screening. A total of 458 articles underwent title and abstract screening and 100 articles underwent full text screening for eligibility; eight articles met the inclusion criteria.

The most common reasons for exclusion was that the study did not use an eligible intervention (N = 288 articles, 63% of screened articles), or that a cRCT study design was not used (N = 252, 55%). Many (N = 301, 66%) articles were excluded for more than one reason.

### Cluster randomised control studies used to investigate interventions other than non-targeted insecticide for reducing mosquito-borne disease in humans

Of the eight included studies outlined in Table 1 and S3 Appendix, five aimed to reduce malaria through targeting *Anopheles* mosquitoes[22–26] and three aimed to reduce dengue by targeting *Aedes aegypti*.[8,27,28] Key characteristics of the included studies are described in Table 1, and are outlined in further detail in S3 Appendix. All eight cRCTs were conducted in low-middle income countries; three studies included urban neighbourhoods in America

**Table 1. Characteristics of included studies.**

| Citation | 1st author, publication year | Location and setting | Disease of interest | Study population | Intervention/s | Targeted mosquito species | Primary disease outcome measure | Effectiveness of intervention |
|---|---|---|---|---|---|---|---|---|
| [27] | Degener C.M. 2014 | Urban neighbourhoods in Manaus, Brazil. | Dengue | 1,487 households with approx. 6,300 inhabitants. | Mass adult mosquito trapping using BG Sentinel traps (approx. 26 traps/hectare); 1 trap per participating (opt-in) household used 24/7 for 17 months. | *Ae. aegypti* | OR of dengue infection using rapid IgM bloodspot tests compared between study arms during the last 2 months of the study period. | OR 2.84 (P = 0.288; Fishers exact test), not statistically significant. |
| [8] | Andersson N. 2015 | Urban and rural areas in Mexico (Guerrero State) and urban areas in Managua, Nicaragua. | Dengue | 9,894 children living in study clusters aged 3–9 years-old. | Chemical-free reduction of mosquito reproduction using approaches tailored to each cluster. Interventions included cleaning interior walls in houses, and covering receptacles used for mosquito breeding. Wastewater clean-up campaigns implemented. Fish introduced to non-drinking water containers in some Mexican clusters. | *Ae. aegypti* | RRR of dengue infection. Dengue IgG detected by paired saliva sampling following the dengue season compared between study arms. | RRR: 29.5% (95% CI: 3.8% to 55.3%), statistically significant. |
| [22] | Syafruddin D. 2014 | Umbungedo village and Wainyapu village (rural), Southwest Sumba District, East Nusa Tenggara Province, Indonesia. | Malaria | 170 malaria free, healthy resident men aged 18–60 years living in separate households and slept in the village >90% of nights, and had no plans for extended travel during the study period. | 4 burning spatial repellent coils in each participant's dwelling. The coils used in the intervention arm were 90% active and 10% placebo type, while coils in the control arm were 90% placebo and 10% active type. | *Anopheles* | Relative risk (RR) of malaria infection detected using weekly blood smear screening for *Plasmodium* parasitaemia compared between study arms. | RR: 0.65 (95% CI: 0.09–4.8), not statistically significant. |
| [28] | Degener C.M. 2015 | Cidade Nova neighbourhood (urban), Manaus, Brazil. | Dengue | 775 households included; 340 participants provided serological samples. | Mass adult mosquito trapping using Sticky Trap MosquiTRAP x3 per participating household (206 households opted in at baseline out of 403 total households in intervention clusters). | *Ae. aegypti* | OR of dengue infection using rapid IgM bloodspot tests compared between study arms during the last 2 months of the study period. | OR 1.08 (P = 1; Fisher's exact test), not statistically significant. |

*(Continued)*

**Table 1.** (Continued)

| Citation | 1st author, publication year | Location and setting | Disease of interest | Study population | Intervention/s | Targeted mosquito species | Primary disease outcome measure | Effectiveness of intervention |
|---|---|---|---|---|---|---|---|---|
| [23] | Yapabandara A.M. 2001 | Rural villages in Kaluganga area, Matale District, Sri Lanka. | Malaria | 4,566–4,659 study area residents. | Periodic application of pyriproxyfen in gem pits and river pools up to 1.5km from intervention villages | *Anopheles culicifacies Anopheles subpictus Anopheles aruna* | RR of malaria case presentation to primary healthcare clinics in the post-intervention year compared between study arms. | RR: 0.24 (95% CI: 0.20–0.29), statistically significant. |
| [24] | Yapabandara A.M. 2004 | Rural villages in Kaluganga area, Matale District, Sri Lanka. | Malaria | Approximately 15,415 study area residents. | Periodic targeted application of pyriproxyfen to riverbeds, streams, irrigation ditches, quarry pits and agricultural wells. | *Anopheles culicifacies Anopheles subpictus* | RR of malaria case presentation to primary healthcare clinics in the post-intervention year compared between study arms. | RR: 0.30 (95% 0.22–0.42), statistically significant. |
| [25] | Sluydts V. 2016 | Rural villages in Ratanakiri Province, Cambodia. | Malaria | 48,838 residents of the most malaria endemic villages which were accessible in the rainy season. | Topical repellent with instructions for daily use provided to all intervention area households. | *Anopheles* | IRR of malaria detected by *Plasmodium* parasitaemia using fingerpick blood screening and real-time PCR analysis. | IRR: 0·94 (95% CI: 0·64–1·39), not statistically significant. |
| [26] | Hill N. 2007 | Rural villages in Vaca Diez and Pando Provinces, and the outer 10% of peri-urban districts around Riberalta and Guayaramerin towns, Bolivia. | Malaria | 4,008 malaria-free residents of households in study area where house was ≥25m from any other participating household. | Eucalyptus-based topical insect repellent with instructions for daily use provided to all people living in intervention households. | *Anopheles darlingi* | IRR of *Plasmodium falciparum* malaria detected using rapid diagnostic tests, adjusted for age, compared between study arms and recorded at monthly follow up surveys. | aIRR: 0.18 (95% CI: 0.02–1.40), not statistically significant. |

BG: Biogents, OR: Odds ratio, RRR: relative risk reduction, IRR: incidence rate ratio, aIRR: adjusted incidence rate ratio, RR: rate ratio, CI: confidence interval, Ig: Antibody.

[8,27,28] while others included villages and outer peri-urban areas in America, South and Southeast Asia. Cluster sizes varied from single households[26] to areas with >1000 inhabitants;[23,24] the number of clusters per study ranged from 4–860.[22,26] The minimum distance between control and intervention cluster residences ranged from around 10m[22] to 3km.[23,24] One study reported outcomes only in children aged 3–9 years[8] and another in men aged 18–60 years;[22] the remaining six studies reported outcomes in all residents of the study area. Baseline prevalence data concerning the disease of interest was reported in six studies,[8,23–26] including one which ensured participants were disease-free at baseline.[22] Five studies stratified clusters at baseline prior to randomisation using a combination of mosquito density, disease incidence and population size measures.[23–25,27,28] Clusters did not appear to be matched in three studies.[8,25,26]

**Table 2. Study strengths and potential limitations.**

| Citation | 1st author, publication year | Study strengths | Study limitations |
|---|---|---|---|
| [27] | Degener C.M. 2014 | • Clusters paired on baseline mosquito density with intervention randomly assigned to one cluster per pair.<br>• Outcomes measured clearly and reliably.<br>• Trial design and statistical methods appear appropriate.<br>• Active case finding. | • Lack of baseline dengue seroprevalence data.<br>• Reduced household intervention participation rates toward end of study (decreased from 60.5% to 36%).<br>• Low dengue transmission during the study period.<br>• The low rate of dengue and high *Culex* catch rates may have led to false reassurance with subsequent relaxing of anti-dengue measures.<br>• Unclear how many households used BG-traps continuously; trapped mosquitoes may have been lost during power cuts and eaten by ants entering the catch bag.<br>• Contamination risk from migrating mosquitoes; minimum distance between clusters of 250 metres. |
| [8] | Andersson N. 2015 | • High community engagement achieved by involving community members, creating community-led campaigns.<br>• Randomisation used to assign intervention.<br>• Outcomes measured clearly and reliably.<br>• Trial design and statistical methods appear appropriate.<br>• Active case finding. | • Non-participation bias among wealthier people.<br>• Security issues reduced intervention participants' engagement with researchers.<br>• Sharing the baseline results with participants in the control and intervention arms may have mobilised both groups to perform anti-dengue control measures.<br>• Entomology evaluators were not blinded.<br>• Intensive government anti-dengue campaigns reduced the difference between study arms.<br>• Two non-participating clusters were included in the intervention arm in the data analysis.<br>• Contamination risk from migrating mosquitoes. |
| [22] | Syafruddin D. 2014 | • Randomisation used to assign intervention.<br>• Outcomes measured clearly and reliably.<br>• Active case finding.<br>• Placebo controlled. | • Four burning spatial repellent coils per house each night was not practical to implement.<br>• The two participating villages (each with an intervention and control arm) had very different baseline malaria rates (heterogeneity).<br>• Few clusters (N = 4 in total).<br>• Possible contamination from mosquitoes in the intervention cluster being diverted to the control cluster in each village, no buffer zones. |
| [28] | Degener C.M. 2015 | • Clusters paired on baseline mosquito density with intervention randomly assigned to one cluster per pair.<br>• Outcomes measured clearly and reliably.<br>• Trial design and statistical methods appear appropriate.<br>• Intervention and control arms were similar at baseline.<br>• Active case finding. | • No buffer zones, small cluster size (possible contamination).<br>• Possibly low mosquito trapping efficiency.<br>• Post intervention increase in mosquito numbers observed with a decrease in mosquito numbers observed in control clusters.<br>• Low participation rate among intervention cluster households.<br>• Possibly too few traps used per intervention house.<br>• Intervention participants may have relaxed anti-dengue measures having felt reassured by the MosquiTRAP traps (false reassurance).<br>• Fewer control participants in serological survey.<br>• Dengue virus IgM not evaluated at baseline.<br>• The study was performed during a time of low dengue transmission. |
| [23] | Yapabandara AM. 2001 | • 1.5km buffer zones between clusters.<br>• Passive case surveillance but access to primary care enhanced.<br>• Clusters stratified on baseline malaria incidence with intervention randomly assigned to half the clusters in each strata.<br>• Outcomes measured clearly and reliably.<br>• Trial design and statistical methods appear appropriate.<br>• High community engagement. | • Contamination risk from migrating mosquitoes and people–highly mobile human population.<br>• Area is not representative of other parts of Sri Lanka. |
| [24] | Yapabandara AM. 2004 | • 1.5km buffer zones between clusters.<br>• Passive case surveillance but access to primary care enhanced.<br>• Clusters stratified on baseline malaria incidence with intervention randomly assigned to half the clusters in each strata.<br>• Outcomes measured clearly and reliably.<br>• Trial design and statistical methods appear appropriate.<br>• High community engagement.<br>• Large study area which was representative of other parts of Sri Lanka. | • Residual house spraying with lambdacyhalothrin occurred in both study arms during the study period (in June and November each year) as part of a government campaign, and this may have diluted the intervention effect.<br>• Pyriproxyfen was not applied to paddy fields.<br>• Contamination risk from migrating mosquitoes and people–highly mobile human population. |

*(Continued)*

**Table 2.** (Continued)

| Citation | 1st author, publication year | Study strengths | Study limitations |
|---|---|---|---|
| [25] | Sluydts V. 2016 | • Clusters stratified by malaria endemicity and population size at baseline with intervention randomly assigned to half the clusters in each strata.<br>• High community engagement.<br>• Active case finding.<br>• Primary outcomes measured clearly and reliably. | • Insufficient statistical power to show an effect from the intervention.<br>• Medical treatment for malaria cases may have reduced case numbers detected at future survey points.<br>• Possible poor compliance with topical repellent use (maybe as low as 15% compliance) in the intervention arm with no direct confirmation of entomological endpoint (reduced blood feeding).<br>• Contamination risk from migrating mosquitoes. |
| [26] | Hill N. 2007 | • Active case finding.<br>• High compliance (98.5%) with the study intervention in the placebo arm and the intervention arm.<br>• Similar loss to follow up across both study arms.<br>• Randomisation used to assign intervention to households.<br>• Outcomes measured clearly and reliably.<br>• Trial design and statistical methods appear appropriate.<br>• High community engagement.<br>• Placebo controlled. | • An unexpected round of outdoor fogging with lambdacyhalothrin was performed by some health districts governments mid-way through the trial, affecting some clusters in both study arms.<br>• Low incidence of *P. falciparum* observed during the study period.<br>• Contamination risk from migrating mosquitoes.<br>• No direct assessment of repellent effectiveness (such as through reduced feeding by mosquitoes) |

A variety of interventions were studied, including: mass adult mosquito trapping to reduce vector abundance;[27,28] topical repellents;[25,26] burning repellent coils at night in participants' homes;[22] pyriproxyfen (a larvicide);[23,24] and community-led source reduction.[8] Six studies involved interventions requiring participant compliance with the intervention procedure, for example by regularly using mosquito traps, repellents, or source reduction measures.[8,22,25–28]

In three studies the intervention resulted in a statistically significant reduction in the disease of interest associated with the intervention.[8,23,24] One large study noted a 30% (95% CI: 4–55%) relative risk reduction in dengue infection among children. This study also observed reductions in entomological indicators of vector breeding (between 35% and 52%; $p < 0.05$) [8] A smaller study noted a relative risk of 0.24 (95% CI: 0.20–0.29) for malaria in the intervention arm compared with the control arm following intervention implementation. Two entomological indicators of breeding for the target vectors also declined (between 50–84%; $p < 0.05$), while non-statistically significant reductions in a third indicator were reported.[23] Another large study noted a relative risk of 0.30 (95% CI: 0.22–0.42) for malaria in the intervention arm compared with the control arm. Reductions (between 72–77%; $p < 0.05$) in an indicator of target vector breeding were noted.[24] Both malaria studies observed corresponding increases in vector breeding indicators in the control arm.[23,24]

Of the five studies which did not observe disease reductions associated with the intervention; two malaria studies did not appear to consider entomological outcome measures,[25,26] and one malaria study observed a 32% reduction in the vector attack rate ($p < 0.05$), however low vector numbers were included and other entomological indicators were not considered. [22] One dengue study did not observe any difference in entomological indicators from mass mosquito trapping[28], while a second, similar study observed a reduction ($p < 0.05$) in vector abundance during the first of three intervention seasons only.[27]

## Strengths and limitations of included study designs

A number of limitations and strengths common among the included studies were apparent (Table 2), including:

**Contamination:** All included studies may have been affected by contamination due to mosquito or person movement between intervention and control cluster areas, and from outside

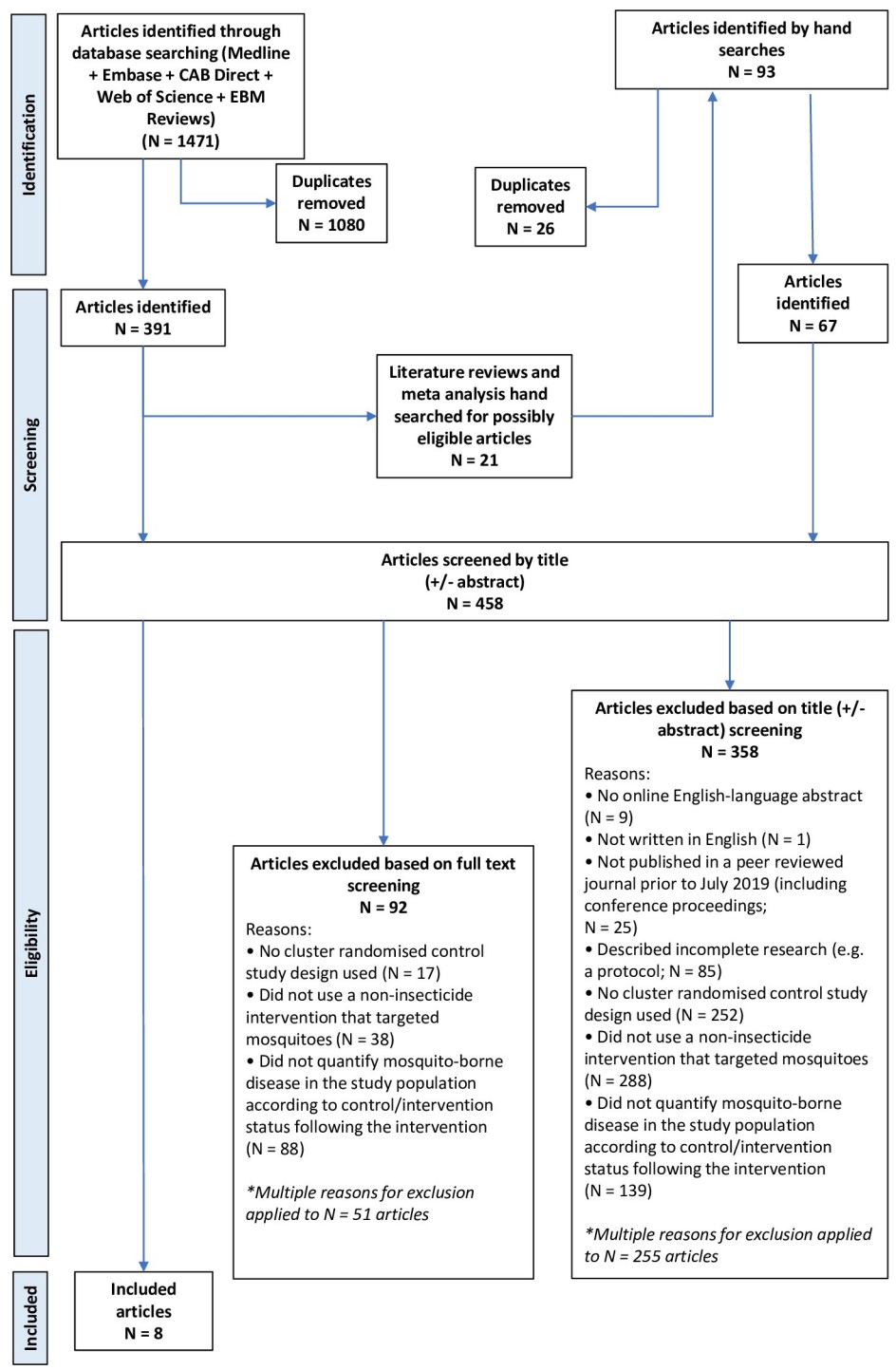

**Fig 1. PRISMA Diagram.**

the study area. Most studies attempted to reduce mosquito contamination by ensuring a minimum distance between clusters, however space between clusters was consistently within the vector species' known flight ranges.[23–28] Potential contamination due to the study population being highly mobile was noted in two studies.[23,24] Three studies reported non-targeted

insecticide application occurred in both intervention and control clusters through government-initiated mosquito control activities. Such applications may have diluted the study intervention effects on the outcome measures.[8,24,26]

**Heterogeneity:** Considerable heterogeneity between study clusters was noted in two studies.[22,28] In one study, only four clusters were used and these were spread across two villages which had very different malaria rates prior to actively clearing malaria infections from participants during the recruitment phase.[22] Heterogeneity in the other study may have been related to differences in cluster ecology. These authors noted a significant vector increase in the intervention arm after initial baseline vector abundance measurements, which was hypothesised to have increased the risk of dengue.[28] Heterogeneity in most included studies was reduced by assessing vector abundance data (a proxy for disease risk) and/or disease rates in baseline clusters, then pairing clusters and randomly assigning the intervention to one cluster in each pair. Despite assessing vector baseline abundance, two studies lacked baseline seroprevalence data for the outcome disease.[27,28]

**Low participant compliance with study intervention:** Possible poor compliance was noted by a study observing topical repellent use for mosquito blood feeding reduction, however the extent of compliance was not well quantified.[25] False reassurance (likely due to the presence of mosquito traps, high catch rates and/or low dengue transmission during the study period) may have led to reduced participant compliance with the intervention in another study, where sub-optimal compliance was also partly attributed to participants' concerns around the cost of electricity to run the mosquito traps and a risk of traps catching fire. Furthermore, the intervention coverage here (i.e. the length of time each household's mosquito trap was used) was not monitored.[28]

**Blinding:** Full blinding was only possible where a placebo was used in the control arm. Placebo repellents were used by two studies, with participants and staff delivering the intervention blinded.[22,26] In one of these studies the outcome assessors were also blinded.[22]

**Statistical under-powering:** Underpowering was noted in four studies, none of which showed a statistically significant effect on the disease outcome.[25–28] Low disease transmission in the study area affected three of these studies.[26–28]

**Community engagement:** Two of the three studies which demonstrated statistically significant disease reductions described high community engagement with the intervention. This resulted in effectively locating and neutralising mosquito breeding sites, and comprehensive case finding.[8,23] In the dengue reduction study,[8] high community engagement with the intervention was achieved through: 1) Requesting permission to conduct the study from community leaders and discussing baseline evidence that the trial could be beneficial; 2) Conducting intervention design groups (usually with 8–10 community members per group) to discuss survey results, cost implications, and specific prevention strategies for each community; 3) Inviting local volunteers to train and work as intervention organisers and educate their communities about source reduction. Similarly, prior to a malaria reduction trial commencing, meetings were held for intervention area residents to discuss potential benefits of the trial and understand the importance of community participation for its success. A volunteer was selected from each group of 20 households. These volunteers helped field staff facilitate community discussion, and located and treated mosquito breeding sites. Community engagement strategies in the third study which demonstrated a statistically significant disease reduction were not described.[24]

**Effective case finding:** In two studies, two new field clinics were set up to diagnose and treat malaria cases in addition to two pre-existing clinics serving the study population. Study area residents were encouraged to attend a clinic if they developed malaria symptoms. Despite using passive surveillance for case finding, both studies showed a statistically significant

malaria reduction in the intervention arm compared with the control arm. [23,24] Active case finding was used in the study which demonstrated a statistically significant dengue reduction following intervention, wherein paired saliva samples were obtained from participants and assayed for dengue antibodies to detect new infections.[8]

### Bias assessment

All included studies were considered to: 1) treat study groups identically, other than by applying the intervention of interest; and 2) report completed participant follow-up or adequately describe and analyse differences between study arms. Participants were consistently analysed in the groups to which they were randomised, with outcomes measured and analysed consistently, reliably and appropriately. The trial designs were largely considered appropriate. Only three studies described the method of randomisation used.[8,27,28] Allocation to treatment groups was concealed in three studies.[8,22,26] Six studies demonstrated the study arms were similar at baseline with regards to the primary outcomes.[8,23,24,26–28]. Blinding occurred in three studies.[8,22,26] Five studies randomly assigned the intervention to one cluster in each cluster pair.[23–25,27,28] Three studies reported on participant loss to follow up.[8,22,26] Bias assessment results are displayed in S3 Appendix.

## Discussion

This review indicates what combinations of mosquito-control inteventions are most effective for reducing mosquito-borne disease in local human populations. Five studies which targeted *Anopheles* species were identified, including two which reduced malaria through widespread larvicide applications.[23,24] Three studies that targeted *Aedes* species were identified, including one which achieved a significant reduction in dengue through mosquito breeding source reduction. Intensive community engagement was common to all three studies with interventions that which successfully led to a reduction in mosquito borne disease.[8,27,28] A key strength of this article lies in the identification of 'real world' effects on key outcome measures, despite the likelihood of contamination between control and intervention arms. These findings can inform future mosquito control activities.

To fully implement a mosquito control intervention and assess the effects on subsequent human disease, community engagement is vital. When mosquito-borne disease rates show considerable year-to-year variation,[16] simply applying an intervention to the entire study area and comparing to pre-baseline rates would not provide evidence of the intervention's effectiveness, although this approach has demonstrated the effectiveness of gravitraps in reducing dengue incidence in Singapore by focussing on multiple sites.[29] Furthermore, a cRCT study design has the advantage of controlling for the movements and behaviours of study area residents, therefore reducing the impact of contamination bias on outcome measures. Due to the limited size of the proposed study area, it may not be possible to incorporate buffer zones between control and intervention clusters to reduce contamination from mosquito movement without compromising statistical power.

Initial community engagement should involve discussing the issue with stakeholders, ascertaining how acceptable proposed interventions are, and tailoring strategies to maximise participant compliance with the intervention. Drawing on local residents' knowledge in all three cRCT interventions that reduced mosquito-borne disease led to effective identification of mosquito breeding sites and comprehensive intervention implementation.[23,24,30] The study which successfully reduced dengue rates utilised the Socialisation of Evidence for Participatory Action (SEPA) approach. Their initial step involved sharing locally relevant scientific evidence with communities and service providers, using "...epidemiology to build the voice of the

communities into planning".[31] Local volunteers were subsequently recruited to conduct house-to-house visits where they discussed and implemented mosquito control activities with consent and involvement from household members. Mosquito control activities were implemented in ingenious ways tailored to suit participating households.[32] Communities exerted considerable control over the intervention and thus were empowered with a sense of ownership of the study. Effective community engagement was observed to promote complete case ascertainment by empowering cases to come to the researchers' attention and become included in the study outcome measures.[23,24,30]

Several important study design considerations were highlighted by this review. First, randomly assigning the intervention to clusters which are stratified and paired according to the baseline disease risk reduces heterogeneity between study arms. Heterogeneity between clusters presented issues in several studies (Table 2). Baseline data should also provide a realistic indication of changes that would occur independently of the intervention during the study period for all major outcome measures. Second, underpowering due to low disease transmission was an important issue which affected several studies.[26–28] Underpowering possibly could have been overcome by extending the intervention and follow-up periods, or expanding the study area. The small number of clusters included in at least one study likely unbalanced the study arms and reduced statistical power to detect changes in outcome measures.[22] Third, the risk of contamination can theoretically be mitigated by incorporating buffer zones around intervention and control areas which exceed the maximum vector flight distance. In reality, incorporating such zones without introducing considerable heterogeneity would be challenging, particularly as *Aedes* species tend to disperse up to 500m from their hatching site [33] and *Anopheles* species can disperse 10km per day.[34] Mosquito-control activities may take place within the study area separately from the study, for example through government-initiated programmes. While matching and randomisation of study clusters may help balance the effects of such activities between study arms, the intervention impact on outcome measures may subsequently be reduced.[8,24,26] Ideally, local government would work with the researchers to ensure public health obligations are met without introducing significant contamination bias. Neither study which used a mass adult mosquito trapping intervention was effective in reducing mosquito-borne disease.[27,28] No data around optimising the intervention impact on mosquito abundance were presented, such as by varying trap density. Mosquito trapping intervention may be particularly vulnerable to contamination (such as though influxes of mosquitoes from non-intervention areas, or people being exposed to mosquitoes outside the intervention area), and poor participant compliance with proposed trapping regimes. However, mosquito trapping has been effective in reducing mosquito abundance in other studies.[29,35,36] The study design should maximise participant compliance with the intervention and assess compliance levels. Measures of self-reported compliance can be complemented by other investigations, such as unannounced checks.[25,26]

Limitations of this systematic review include the small number of eligible studies (N = 8) and the wide range in study settings and target vector species. As these studies are spread across different continents, populations and vectors, the results from one study (or even from a single cluster) are not necessarily generalisable to other areas. Eligible cRCTs addressing mosquito-borne diseases other than malaria or dengue were not identified, so how well these findings can be applied to other conditions is unknown. Interventions considered by this review were limited to source reduction, larviciding using insect growth regulators, adult mosquito trapping and repellent use. As we searched five electronic databases and reference lists of identified review articles, it is unlikely eligible published articles were missed. When applying the bias assessment tool to included studies, it was sometimes unclear in the article how well the study fulfilled some of the criterions (S3 Appendix). The use of this tool does not preclude

the need for careful reading, critical appraisal and clinical reasoning when considering whether to include data from flawed studies.

Our extensive literature search did not identify any cRCTs aiming to disrupt mosquito transmission of Buruli ulcer, or targeting *Aedes notoscriptus* (a suspected Buruli ulcer vector). This emphasises the novel nature of the proposed Buruli ulcer prevention strategy. Our findings tentatively support the use of a cRCT with a source reduction intervention strategy (incorporating larvicide) targeting *Aedes* species to reduce endemic *Aedes*-transmitted disease if effective community engagement can be achieved. The findings reflect that of Alvarado-Castro *et al*., in their review investigating the effects of control measures on *Ae. aegypti* proliferation. The authors observed that community mobilisation was consistently effective while effects of chemical and biological control interventions were more mixed, although the effects on dengue were not considered.[9] When designing a cRCT to disrupt mosquito-borne disease transmission, gathering appropriate baseline data on the vector abundance and disease rates is imperative for assessing the impact of the intervention, as well as ensuring that intervention/s have a substantial impact on mosquito abundance and biting frequency. Robust baseline data should enable clusters to be paired and stratified by disease risk, with the intervention then randomly allocated to one cluster from each pair. Mechanisms to assess impact of non-study activities by participants or communities on outcome measures should be incorporated into the study design, such as by assessing the prevalence and types of household- and local government-initiated mosquito controls during the study period. High community support for, and engagement with, mosquito control interventions, are essential for success.

## Supporting information

**S1 Appendix. Search Strategy.**
(XLSX)

**S2 Appendix. Data fields extracted.**
(DOCX)

**S3 Appendix. Included Articles—data extraction template.**
(XLSX)

**S1 PRISMA Checklist. PRISMA 2009 Checklist.**
(DOC)

## Author Contributions

**Conceptualization:** Simon Crouch.

**Data curation:** Jane Oliver, Stuart Larsen.

**Formal analysis:** Jane Oliver, Stuart Larsen.

**Investigation:** Jane Oliver.

**Methodology:** Jane Oliver, Ary Hoffmann, Simon Crouch, Katherine B. Gibney.

**Project administration:** Jane Oliver, Tim P. Stinear.

**Resources:** Jane Oliver, Stuart Larsen.

**Software:** Jane Oliver, Stuart Larsen.

**Supervision:** Simon Crouch, Katherine B. Gibney.

**Validation:** Jane Oliver, Stuart Larsen.

**Visualization:** Jane Oliver, Stuart Larsen.

**Writing – original draft:** Jane Oliver, Simon Crouch.

**Writing – review & editing:** Jane Oliver, Stuart Larsen, Tim P. Stinear, Ary Hoffmann, Simon Crouch, Katherine B. Gibney.

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
