## [Decision Letter · Decision Letter 0]

21 Mar 2021

Dear Dr Oliver,

Thank you very much for submitting your manuscript "Cluster randomised controlled studies to investigate interrupting mosquito-borne disease transmission to humans without non-targeted insecticide A systematic review" for consideration at PLOS Neglected Tropical Diseases. As with all papers reviewed by the journal, your manuscript was reviewed by members of the editorial board and by several independent reviewers. In light of the reviews (below this email), we would like to invite the resubmission of a significantly-revised version that takes into account the reviewers' comments. 

Reviewer #1: PLOS NTD

Title : Cluster randomised controlled studies to investigate interrupting mosquito-borne disease transmission to humans without non-targeted insecticide A systematic review

The authors have included 2 studies where non-targeted insecticide was applied in both arms.

It is not clear that the included studies that "interrupt" transmission - I suggest "reduce".

Also this review has been conducted to inform the use of a cRCT to reduce mosquito borne transmission of Mycobacterium ulcerans, the causal agent of Bureli ulcer. "A novel intervention study for Buruli ulcer targeting mosquito vectors was proposed for a Buruli ulcer-endemic area of Victoria, Australia"

Thus a more infomative title for this article: 

Cluster randomised controlled studies to reduce mosquito-borne disease transmission to humans. A systematic review in the context of Bureli ulcer transmission.

Support for the evidence that mosquitoes transmit Bureli ulcer, appears to be weak. Page 3, second paragraph "Mycobacterium ulcerans is considered an environmental pathogen…., evidence implicates biting insects as having a key role in the transmission of M. ulcerans. [16, 17]." Thus in their proposed intervention it is important that the methods used to reduce mosquitoes density are highly efficacious. 

Abstract: 

State the date range applied to interrogate the listed medical research databases.

Needs more care with English, for example in the abstract, results section:

"Eight eligible cRCTs were conducted between 1994-2013 in variable settings in the Americas and Asia were identified." Suggestion "Eight eligible cRCTs conducted between 1994-2013 were identified in a variety of settings in the Americas and Asia."

The term "Source reduction" requires definition. 

Page 6 

Data abstraction: The full …….disease and vector; study setting and period;. Please change "period" to "date" 

The aim was to review cRCT designs used to investigate interventions without non-targeted insecticide for reducing mosquito-borne disease transmission to humans, and comment on the

strengths and weaknesses of these study designs. They have included articles with non-targeted insecticide when applied in control and intervention groups. This needs to be justified.

DISCUSSION - first sentence:

"This review is unique in that it considers the effects of mosquito control interventions on associated disease, rather than on entomological indicators only" 

Please clarify. Many studies and reviews consider the effect of mosquito control interventions on associated disease - Anopheline studies and malaria and many others.

The novel nature of the proposed Buruli ulcer prevention strategy is, I believe, that mosquitoes have not previously been targeted as a means of reducing this disease.

Reviewer #2: Well written paper. Considering that the review include small number of eligible studies, it would be helpful to discuss advantages and disadvantages of other types of study design used for similar studies

We cannot make any decision about publication until we have seen the revised manuscript and your response to the reviewers' comments. Your revised manuscript is also likely to be sent to reviewers for further evaluation.

Sincerely,

Elvina Viennet, PhD

Deputy Editor

Elvina Viennet

Deputy Editor

Reviewer #1: PLOS NTD

Title : Cluster randomised controlled studies to investigate interrupting mosquito-borne disease transmission to humans without non-targeted insecticide A systematic review

The authors have included 2 studies where non-targeted insecticide was applied in both arms.

It is not clear that the included studies that "interrupt" transmission - I suggest "reduce".

Also this review has been conducted to inform the use of a cRCT to reduce mosquito borne transmission of Mycobacterium ulcerans, the causal agent of Bureli ulcer. "A novel intervention study for Buruli ulcer targeting mosquito vectors was proposed for a Buruli ulcer-endemic area of Victoria, Australia"

Thus a more infomative title for this article: 

Cluster randomised controlled studies to reduce mosquito-borne disease transmission to humans. A systematic review in the context of Bureli ulcer transmission.

Support for the evidence that mosquitoes transmit Bureli ulcer, appears to be weak. Page 3, second paragraph "Mycobacterium ulcerans is considered an environmental pathogen…., evidence implicates biting insects as having a key role in the transmission of M. ulcerans. [16, 17]." Thus in their proposed intervention it is important that the methods used to reduce mosquitoes density are highly efficacious. 

Abstract: 

State the date range applied to interrogate the listed medical research databases.

Needs more care with English, for example in the abstract, results section:

"Eight eligible cRCTs were conducted between 1994-2013 in variable settings in the Americas and Asia were identified." Suggestion "Eight eligible cRCTs conducted between 1994-2013 were identified in a variety of settings in the Americas and Asia."

The term "Source reduction" requires definition. 

Page 6 

Data abstraction: The full …….disease and vector; study setting and period;. Please change "period" to "date" 

The aim was to review cRCT designs used to investigate interventions without non-targeted insecticide for reducing mosquito-borne disease transmission to humans, and comment on the

strengths and weaknesses of these study designs. They have included articles with non-targeted insecticide when applied in control and intervention groups. This needs to be justified.

DISCUSSION - first sentence:

"This review is unique in that it considers the effects of mosquito control interventions on associated disease, rather than on entomological indicators only" 

Please clarify. Many studies and reviews consider the effect of mosquito control interventions on associated disease - Anopheline studies and malaria and many others.

The novel nature of the proposed Buruli ulcer prevention strategy is, I believe, that mosquitoes have not previously been targeted as a means of reducing this disease.

Reviewer #2: Well written paper. Considering that the review include small number of eligible studies, it would be helpful to discuss advantages and disadvantages of other types of study design used for similar studies
---

## [Decision Letter · Decision Letter 1]

9 May 2021

Dear Dr Oliver,

Thank you very much for submitting your manuscript "Reducing mosquito-borne disease transmission to humans: A systematic review of cluster randomised controlled studies that assess interventions other than non-targeted insecticide" for consideration at PLOS Neglected Tropical Diseases. As with all papers reviewed by the journal, your manuscript was reviewed by members of the editorial board and by several independent reviewers. In light of the reviews (below this email), we would like to invite the resubmission of a significantly-revised version that takes into account the reviewers' comments. 

Reviewer #1: Overall

This review presents a systematic review of published cluster randomised control studies (cRCT) of mosquito control interventions including breeding source reduction, using mosquito-borne disease as an outcome. A previous review has used entomological endpoints (Alvarado-Castro V, et al. ,2017, cited at the end of the article) but I believe that use of a clinical endpoint is both novel and useful. 

The authors state the review was conducted to inform the design of a future cRCT for Buruli ulcer prevention in Victoria. Strictly limited details of the planned Buruli ulcer cRCT add interest to this article but should be succinct. Similar considerations apply to the Discussion.

The article thus requires major revision.

The English also needs revision.

In addition, given that only three articles have been identified by this systematic review, perhaps the article would be more suitable as a short communication. 

An additional strength of this article is Table 2. These are real world results that can inform future studies.

The phrase "interventions with targeted insecticide" is considerably clearer than "interventions without nontargeted insecticide".

Specific comments

Author Summary

Point 2 This literature review identified three studies with an intervention which did not include non-targeted use of insecticide and was associated with statistically significant reductions in the disease of interest and in entomological indicators.

Rewrite: This literature review identified three intervention studies, without non-targeted use of insecticide, that were associated with statistically significant reductions.

Point 3 & 4 are repetitive and should be combined.

Point 3 High community engagement is vital for the success of a cluster randomised control study aiming to reduce mosquito-borne disease with a mosquito control intervention.

Point4 A mosquito breeding source reduction intervention for Aedes control may effectively reduce disease transmitted by this vector in endemic areas if local communities are supportive and very engaged.

Point 5 Is the final statement supported by the evidence reviewed? "Regular administration of larvicide to potential breeding sites that are unsuitable for source reduction may supplement this intervention strategy." If not, I think it should be removed.

Abstract

Well written

Title

Full Title: "Reducing mosquito-borne disease transmission to humans: A systematic review of cluster randomised controlled studies that assess interventions other than non-targeted insecticide."

Suggest: "Reducing mosquito-borne disease transmission to humans: A systematic review of cluster randomised controlled studies that assess interventions using targeted insecticide."

Short title: Intervention studies without non-targeted insecticide to reduce mosquito-borne disease.

Suggest: "cRCT using targeted insecticide to reduce mosquito-borne disease."

Introduction

Section on Buruli ulcer only need mention:

"This literature review was developed to inform the design of a future cluster randomised control study (cRCT) aiming to reduce Buruli ulcer transmission via a mosquito control intervention that does not include non-targeted insecticide spraying. The aim was to review cRCT designs used to investigate interventions without non-targeted insecticide for reducing mosquito-borne disease transmission to humans, and comment on the strengths and weaknesses of these study designs." 

Methods

The authors definition of studies eligible for this review Page is suitable. The identification and selection of articles is well described, follows standard procedures, and covers "from the earliest available sources" up to July 2019. There are 8 articles included in the review but in the final analysis only three met their criteria.

They quote a previous review of cRDT in this field (Alvarado-Castro V, et al. (2017), ref 41). It would be informative to include this in the introduction as an example of a review on entomological endpoints and highlight reasons for using disease endpoints. 

The inclusion criteria for studies in the review needs clarification. 

Page 6, top paragraph. This indicates that studies with non-targeted insecticide/s applied in both the control and intervention arms were included. 

Page 6, last paragraph the Eligibility criteria states "Intervention targeted mosquitoes without non-targeted use of insecticide."

Page 13, Contamination between clusters probably occurred in several of the studies and are listed in Table 2. These are real world results that can inform future studies and this is a strength of this article.

Results

Page 8, top 

Clarify the meaning of "title/abstract" as Title and abstract, Title or abstract etc

Page 8, top 

Following searches of the five medical research databases, 1,471 article citations were identified of which 391 articles underwent title/abstract screening (with 1,080 citations identified as duplicates and discarded; Figure 1).

Rewrite: Following searches of the five medical research databases, 1,471 article citations were identified of which 1080 were duplicates and 391 articles underwent title/abstract screening (Figure 1). 

Discussion

As mentioned previously, strictly limited / succinct details of the planned Buruli ulcer cRCT add interest to this article. The Discussion needs to discuss the results of the literature review with minor mention of the Buruli ulcer cRCT. It therefore needs rewritting.

We cannot make any decision about publication until we have seen the revised manuscript and your response to the reviewers' comments. Your revised manuscript is also likely to be sent to reviewers for further evaluation.

Sincerely,

Elvina Viennet, PhD

Deputy Editor

Reviewer #1: Overall

This review presents a systematic review of published cluster randomised control studies (cRCT) of mosquito control interventions including breeding source reduction, using mosquito-borne disease as an outcome. A previous review has used entomological endpoints (Alvarado-Castro V, et al. ,2017, cited at the end of the article) but I believe that use of a clinical endpoint is both novel and useful. 

The authors state the review was conducted to inform the design of a future cRCT for Buruli ulcer prevention in Victoria. Strictly limited details of the planned Buruli ulcer cRCT add interest to this article but should be succinct. Similar considerations apply to the Discussion.

The article thus requires major revision.

The English also needs revision.

In addition, given that only three articles have been identified by this systematic review, perhaps the article would be more suitable as a short communication. 

An additional strength of this article is Table 2. These are real world results that can inform future studies.

The phrase "interventions with targeted insecticide" is considerably clearer than "interventions without nontargeted insecticide".

Specific comments

Author Summary

Point 2 This literature review identified three studies with an intervention which did not include non-targeted use of insecticide and was associated with statistically significant reductions in the disease of interest and in entomological indicators.

Rewrite: This literature review identified three intervention studies, without non-targeted use of insecticide, that were associated with statistically significant reductions.

Point 3 & 4 are repetitive and should be combined.

Point 3 High community engagement is vital for the success of a cluster randomised control study aiming to reduce mosquito-borne disease with a mosquito control intervention.

Point4 A mosquito breeding source reduction intervention for Aedes control may effectively reduce disease transmitted by this vector in endemic areas if local communities are supportive and very engaged.

Point 5 Is the final statement supported by the evidence reviewed? "Regular administration of larvicide to potential breeding sites that are unsuitable for source reduction may supplement this intervention strategy." If not, I think it should be removed.

Abstract

Well written

Title

Full Title: "Reducing mosquito-borne disease transmission to humans: A systematic review of cluster randomised controlled studies that assess interventions other than non-targeted insecticide."

Suggest: "Reducing mosquito-borne disease transmission to humans: A systematic review of cluster randomised controlled studies that assess interventions using targeted insecticide."

Short title: Intervention studies without non-targeted insecticide to reduce mosquito-borne disease.

Suggest: "cRCT using targeted insecticide to reduce mosquito-borne disease."

Introduction

Section on Buruli ulcer only need mention:

"This literature review was developed to inform the design of a future cluster randomised control study (cRCT) aiming to reduce Buruli ulcer transmission via a mosquito control intervention that does not include non-targeted insecticide spraying. The aim was to review cRCT designs used to investigate interventions without non-targeted insecticide for reducing mosquito-borne disease transmission to humans, and comment on the strengths and weaknesses of these study designs." 

Methods

The authors definition of studies eligible for this review Page is suitable. The identification and selection of articles is well described, follows standard procedures, and covers "from the earliest available sources" up to July 2019. There are 8 articles included in the review but in the final analysis only three met their criteria.

They quote a previous review of cRDT in this field (Alvarado-Castro V, et al. (2017), ref 41). It would be informative to include this in the introduction as an example of a review on entomological endpoints and highlight reasons for using disease endpoints. 

The inclusion criteria for studies in the review needs clarification. 

Page 6, top paragraph. This indicates that studies with non-targeted insecticide/s applied in both the control and intervention arms were included. 

Page 6, last paragraph the Eligibility criteria states "Intervention targeted mosquitoes without non-targeted use of insecticide."

Page 13, Contamination between clusters probably occurred in several of the studies and are listed in Table 2. These are real world results that can inform future studies and this is a strength of this article.

Results

Page 8, top 

Clarify the meaning of "title/abstract" as Title and abstract, Title or abstract etc

Page 8, top 

Following searches of the five medical research databases, 1,471 article citations were identified of which 391 articles underwent title/abstract screening (with 1,080 citations identified as duplicates and discarded; Figure 1).

Rewrite: Following searches of the five medical research databases, 1,471 article citations were identified of which 1080 were duplicates and 391 articles underwent title/abstract screening (Figure 1). 

Discussion

As mentioned previously, strictly limited / succint details of the planned Buruli ulcer cRCT add interest to this article. The Discussion needs to discuss the results of the literature review with minor mention of the Buruli ulcer cRCT. It therefore needs rewritting.
---

## [Editor Report · Decision Letter 2]

20 Jun 2021

Dear Dr Oliver,

Thank you very much for submitting your manuscript "Reducing mosquito-borne disease transmission to humans: A systematic review of cluster randomised controlled studies that assess interventions other than non-targeted insecticide" for consideration at PLOS Neglected Tropical Diseases. As with all papers reviewed by the journal, your manuscript was reviewed by members of the editorial board and by several independent reviewers. In light of the reviews (below this email), we would like to invite the resubmission of a significantly-revised version that takes into account the reviewers' comments. 

Dear Dr Jane Oliver,

I couldn't find anywhere your response to reviewer point by point. 

Reviewer 1 had interesting comments within the title, Introduction, Methods, Results and Discussion.

If you disagree with the comments and suggestions, please justify.

At present, none of them have been addressed in your Response to reviewers (17 June 2021).

Could you, please, upload a revised response with detailed action?

Thank you in advance.

We cannot make any decision about publication until we have seen the revised manuscript and your response to the reviewers' comments. Your revised manuscript is also likely to be sent to reviewers for further evaluation.

Sincerely,

Elvina Viennet, PhD

Deputy Editor

Reviewer 1 comments:

Overall

This review presents a systematic review of published cluster randomised control studies (cRCT) of mosquito control interventions including breeding source reduction, using mosquito-borne disease as an outcome. A previous review has used entomological endpoints (Alvarado-Castro V, et al. ,2017, cited at the end of the article) but I believe that use of a clinical endpoint is both novel and useful.

The authors state the review was conducted to inform the design of a future cRCT for Buruli ulcer prevention in Victoria. Strictly limited details of the planned Buruli ulcer cRCT add interest to this article but should be succinct. Similar considerations apply to the Discussion.

The article thus requires major revision.

The English also needs revision.

In addition, given that only three articles have been identified by this systematic review, perhaps the article would be more suitable as a short communication.

An additional strength of this article is Table 2. These are real world results that can inform future studies.

The phrase "interventions with targeted insecticide" is considerably clearer than "interventions without nontargeted insecticide".

Specific comments

Author Summary

Point 2 This literature review identified three studies with an intervention which did not include non-targeted use of insecticide and was associated with statistically significant reductions in the disease of interest and in entomological indicators.

Rewrite: This literature review identified three intervention studies, without non-targeted use of insecticide, that were associated with statistically significant reductions.

Point 3 & 4 are repetitive and should be combined.

Point 3 High community engagement is vital for the success of a cluster randomised control study aiming to reduce mosquito-borne disease with a mosquito control intervention.

Point4 A mosquito breeding source reduction intervention for Aedes control may effectively reduce disease transmitted by this vector in endemic areas if local communities are supportive and very engaged.

Point 5 Is the final statement supported by the evidence reviewed? "Regular administration of larvicide to potential breeding sites that are unsuitable for source reduction may supplement this intervention strategy." If not, I think it should be removed.

Abstract

Well written

Title

Full Title: "Reducing mosquito-borne disease transmission to humans: A systematic review of cluster randomised controlled studies that assess interventions other than non-targeted insecticide."

Suggest: "Reducing mosquito-borne disease transmission to humans: A systematic review of cluster randomised controlled studies that assess interventions using targeted insecticide."

Short title: Intervention studies without non-targeted insecticide to reduce mosquito-borne disease.

Suggest: "cRCT using targeted insecticide to reduce mosquito-borne disease."

Introduction

Section on Buruli ulcer only need mention:

"This literature review was developed to inform the design of a future cluster randomised control study (cRCT) aiming to reduce Buruli ulcer transmission via a mosquito control intervention that does not include non-targeted insecticide spraying. The aim was to review cRCT designs used to investigate interventions without non-targeted insecticide for reducing mosquito-borne disease transmission to humans, and comment on the strengths and weaknesses of these study designs." 

Methods

The authors definition of studies eligible for this review Page is suitable. The identification and selection of articles is well described, follows standard procedures, and covers "from the earliest available sources" up to July 2019. There are 8 articles included in the review but in the final analysis only three met their criteria.

They quote a previous review of cRDT in this field (Alvarado-Castro V, et al. (2017), ref 41). It would be informative to include this in the introduction as an example of a review on entomological endpoints and highlight reasons for using disease endpoints. 

The inclusion criteria for studies in the review needs clarification. 

Page 6, top paragraph. This indicates that studies with non-targeted insecticide/s applied in both the control and intervention arms were included.

Page 6, last paragraph the Eligibility criteria states "Intervention targeted mosquitoes without non-targeted use of insecticide."

Page 13, Contamination between clusters probably occurred in several of the studies and are listed in Table 2. These are real world results that can inform future studies and this is a strength of this article.

Results

Page 8, top

Clarify the meaning of "title/abstract" as Title and abstract, Title or abstract etc

Page 8, top

Following searches of the five medical research databases, 1,471 article citations were identified of which 391 articles underwent title/abstract screening (with 1,080 citations identified as duplicates and discarded; Figure 1).

Rewrite: Following searches of the five medical research databases, 1,471 article citations were identified of which 1080 were duplicates and 391 articles underwent title/abstract screening (Figure 1).

Discussion

As mentioned previously, strictly limited / succint details of the planned Buruli ulcer cRCT add interest to this article. The Discussion needs to discuss the results of the literature review with minor mention of the Buruli ulcer cRCT. It therefore needs rewritting.
---

## [Editor Report · Decision Letter 3]

29 Jun 2021

Dear Dr Oliver,

We are pleased to inform you that your manuscript 'Reducing mosquito-borne disease transmission to humans: A systematic review of cluster randomised controlled studies that assess interventions other than non-targeted insecticide' has been provisionally accepted for publication in PLOS Neglected Tropical Diseases.

Best regards,

Elvina Viennet, PhD

Deputy Editor

---

## [Editor Report · Acceptance letter]

23 Jul 2021

Dear Dr Oliver,

We are delighted to inform you that your manuscript, "Reducing mosquito-borne disease transmission to humans: A systematic review of cluster randomised controlled studies that assess interventions other than non-targeted insecticide," has been formally accepted for publication in PLOS Neglected Tropical Diseases.

Best regards,

Shaden Kamhawi

co-Editor-in-Chief

Paul Brindley

co-Editor-in-Chief
